# Effects of CPAP on Language Recovery in Post-Stroke Aphasia: A Review of Recent Literature

**DOI:** 10.3390/brainsci12030379

**Published:** 2022-03-12

**Authors:** Eve Mercer, Eleanor Sherfey, Candice Ogbu, Ellyn A. Riley

**Affiliations:** Aphasia Research Laboratory, Department of Communication Sciences & Disorders, Syracuse University, Syracuse, NY 13244, USA; esherfey@syr.edu (E.S.); akogbu@syr.edu (C.O.)

**Keywords:** stroke, sleep, sleep disorders, CPAP, language, cognition, attention, memory

## Abstract

Obstructive sleep apnea is highly prevalent in the post-stroke population, and has been shown to affect cognitive, neurological, and functional status. Continuous positive airway pressure (CPAP) treatment is one of the most effective interventions for obstructive sleep apnea, but compliance is often low due to confounding effects of co-occurring conditions, side effects of treatment titration procedures, and individual patient personality characteristics, perceptions, and social factors. Current research suggests that CPAP treatment for obstructive sleep apnea is not associated with significant risk and can subsequently improve post-stroke motor and neurocognitive function. However, effects of CPAP treatment on post-stroke speech and language recovery remain unclear. Post-stroke communication disorders (e.g., aphasia, dysarthria, and apraxia) are also highly prevalent in this population. Knowledge of the potential positive impact of CPAP on language recovery could contribute to patients’ motivation to comply with CPAP treatment and provide incentive for speech-language pathologists to refer patients to sleep medicine specialists. In this review of the literature, we examine the question of what effect CPAP treatment may have on post-stroke speech and language function and recovery, as well as summarize the current knowledge on cognitive, neurological, and functional effects. While this review of the literature found CPAP to have varying effects on different cognitive domains, there was not sufficient evidence to determine effects on language recovery. Further research is necessary to determine the potential effects of CPAP treatment on speech and language recovery among stroke patients.

## 1. Introduction

Sleep disordered breathing (SDB) is a set of disorders primarily characterized by respiratory abnormalities during sleep. Defined as an SDB, obstructive sleep apnea (OSA) consists of obstructions in the upper airways and cessations in breathing during sleep.

Among the general population, SDB ranging from moderate to severe is present in about 26% of people [1]. However, the prevalence of moderate to severe SDB in the acute post-stroke population ranges from 50–70% [1]. SBD tends to improve after the initial acute stroke phase; however, by about 3 months post-stroke or transient ischemic attack (TIA), around 60% of individuals exhibit mild sleep apnea, and around 30% have severe sleep apnea [1]. Further, SDB has been found to impact cognitive and functional recovery in stroke patients [2]. When stroke patients experience SDB, it can cause neurological deterioration, lead to longer hospital stays, impact post-stroke mortality rates, negatively impact functional level, and influence both short- and long-term stroke outcomes [2,3].

Continuous positive airway pressure (CPAP) is one of the most effective treatments for OSA and has been shown to improve sleepiness and cognitive performance in the general population [1]. CPAP negates troubled breathing by forcing oxygen into the upper airway and preventing the obstruction of the airway while patients are asleep. However, data on CPAP’s effects on neurologic recovery after stroke are still conflicting. Although identifying and treating OSA in stroke patients was added to AHA/ASA stroke guidelines in 2014, treatment of SDB is still often neglected in this population due to concerns of feasibility and the acute efficacy of CPAP [1]. Despite these clinical limitations, researchers continue to investigate the possible uses of CPAP in the post-stroke population.

A systematic review and meta-analysis by Brill et al. (2018) included 10 randomized control trials (RCTs) that compared the neurological outcomes of stroke or TIA patients treated with CPAP to control patients treated with standard care or sham CPAP [1]. Changes in neurofunction pre- and/or post-treatment were included in six of the ten included studies [1]. The studies analyzed by Brill and colleagues averaged a CPAP adherence of 4.5 h per night, which is fortunately high among this population [1]. CPAP compliance is a common barrier to treatment success, especially among patients with neurological impairment [1]. The results indicated a positive change in neurofunctional status in patients treated with CPAP as compared to controls; however, there was a great amount of variability between studies [1].

Considering the evidence that CPAP treatment may improve cognitive functioning in the post-stroke population, the objective of this review is to further analyze potential neurological effects in this population by including the comorbidity of aphasia. We aim to further understand the link of obstructive SDB and aphasia as co-occurring conditions to potentially utilize CPAP as a multidimensional treatment. Using Brill et al.’s 2018 article as a starting point, we will provide an updated review on the use of CPAP in post-stroke patients to improve neurofunctional outcomes. Additionally, we will discuss the results of the recent literature to assess the potential effects of CPAP on language recovery.

## 2. Methods

### 2.1. Literature Search, Study Selection, and Data Extraction

We performed a systematic search of Medline, Embase, and Cochrane Library, with a date range of 2011 to 2021, to identify studies to include in this review. We chose to use the same databases used in the search by Brill et al. The list of search terms is detailed in Appendix A. Figure 1 outlines the retrieval of studies over the course of the search.

Two undergraduate students (Authors E.S. and C.O.) completed the search. For each database search, the search authors completed 14 specific queries and applied filters (see Appendix A). Results of each specific query contained large numbers of duplicates. In the initial screening phase, authors removed duplicates across databases and queries, removed non-full-text articles (e.g., abstracts, posters, presentations), and removed articles that were not available in English (screening phase in Figure 1). The remaining 74 articles were then screened by title and abstract for relevant terms (continued screening phase in Figure 1). A more thorough evaluation was then performed to assess eligibility according to inclusion criteria (eligibility phase in Figure 1). Exclusion criteria were determined throughout the process of reading articles and evaluating the ability to compare study objectives and outcome measures. Inclusion and exclusion criteria can be found in Table 1. If a consensus between the two undergraduate investigators could not be reached, the principal investigator (E.R.) made the final decision. Annotated bibliographies were created to extract relevant information to be discussed in this review. Three studies met inclusion criteria to be reviewed in the result synthesis, and the remaining articles were excluded based on our criteria in Table 1.

### 2.2. Inclusion and Exclusion Criteria

Studies included in this review met inclusion criteria as follows: were RCTs, were published between 2011 and 2021, and studied CPAP, nCPAP (nasal continuous positive airway pressure), or aCPAP (auto-adjusting continuous positive airway pressure) intervention compared to a control group receiving sham CPAP or TAU. The participants were adults (18+) that had experienced a stroke or TIA and subsequently been diagnosed with SDB. Outcome measures were baseline vs. post-intervention measures of general cognitive or neurological functioning as well as language abilities.

Excluded studies included study designs other than RCTs as well as studies that utilized biPAP (bilevel positive airway pressure), EPAP (expiratory positive airway pressure), or another type of positive airway pressure treatment. Outcome measures related to cardiovascular events, physical functioning, major adverse events (MAEs), or adherence to CPAP treatment were deemed unnecessary to include for the purpose of this review. Any articles with relevant measures but that did not include both baseline and post-intervention measures were also excluded. Articles that were unable to be accessed through inter-library loan (*n* = 2) were additionally excluded due to access issues [4,5]. Duplicate publications found between the databases searched were pruned to a single copy.

### 2.3. Outcome Measures

The primary outcome measures for this review included neurocognitive and language parameters. Secondary outcomes were stroke severity and functional status, sleepiness, and the affective components of depression and anxiety. Any neuropsychological tests used in the included studies were considered valuable, regardless of the level of sensitivity.

### 2.4. Operational Definitions of Values

*p*-values less than 0.05 (*p* < 0.05) were considered significant and were used to compare outcomes between studies.

## 3. Results

### 3.1. Search Results

From the search criteria detailed above, 1376 articles were identified. After duplicates were removed and inclusion and exclusion criteria were applied, three RCTs were identified and included in this review. Descriptive data from each of these three studies are presented in Table 2.

### 3.2. Study Characteristics

The included studies, by Kim et al., Aaronson et al., and Ryan et al., investigated the use of CPAP treatment during rehabilitation of stroke patients to determine its potential benefit for improving function, cognition, language, quality of sleep, sleepiness, and affect [2,3,6]. The investigators for these studies recruited acute stroke patients with SDB and compared a group receiving CPAP to a group receiving the standard course of treatment [2,3,6]. Kim and colleagues used a 3-week period for CPAP treatment, while the remaining two studies used an intervention period of 4 weeks [2,3,6]. Outcome measures were compared from baseline to post-treatment in each study [2,3,6].

### 3.3. CPAP Compliance

As noted by Brill and colleagues, CPAP compliance is important to the success of the treatment, and all studies included in this review echoed that sentiment [1,2,3,6]. However, definitions of compliance and methods of measuring compliance varied from study to study. The studies by Kim et al. and Aaronson et al. defined highly compliant CPAP use as more than 4 h per day for 5 or more days per week [2,3], while Ryan et al. did not specifically state criteria for CPAP compliance [6]. Kim et al.’s study design used observations of CPAP use to report compliance [2], while the studies by Aaronson et al. and Ryan et al. directly measured CPAP compliance by recording the amount of time a participant spent with the mask on, as recorded by the CPAP machine [3,6].

Ryan et al. reported excellent compliance, with an average use of 4.96 h/day [6]. Kim and colleagues also indicated that high treatment compliance was met; however, compliance with CPAP was not measured accurately in their investigation [2]. The CPAP group in the study by Aaronson et al. did not meet compliance (mean compliance = 2.5 h/night) [3].

### 3.4. Primary Outcome Measure: Cognitive Domains

#### 3.4.1. General Assessment of Neurocognition

The study by Kim et al. used the Korean version of the Mini-Mental State Exam (K-MMSE) to measure cognitive improvement [2]. After the intervention period, the group that received CPAP treatment demonstrated significantly more improvement in the cognitive domain as compared to the control group (*p* = 0.045) [2]. The improvement in this measure was also found to correlate with a positive change in the apnea–hypopnea index (AHI) (*p* = 0.033), an objective measure of SDB severity [2].

The Canadian Neurological Scale was used by Ryan et al. to assess cognition [6]. The group that received CPAP treatment demonstrated significant improvements in the cognitive component of the Canadian Neurological scale following treatment (*p* < 0.001), and these improvements were significantly larger compared to the control group (*p* < 0.001) [6]. The authors additionally assessed social cognition (including a component of communication) using the cognitive Functional Independence Measure (FIM) [6]. Within the CPAP group, there was significant improvement in the cognitive FIM (*p* = 0.03) following intervention [6]. However, the between-group difference in outcomes was not significant (*p* = 0.76) [6].

In the study completed by Aaronson and colleagues, there was no overall measure of cognitive ability reported in the outcome measures. Instead, the authors individually assessed various domains of cognition, which will be discussed in subsequent sections.

#### 3.4.2. Vigilance and Attention

Two of the three included studies reported significant improvements in the domains of vigilance and/or attention. In the study by Kim et al., the K-MMSE was used to report a score for “Attention and Calculation” [2]. This measure demonstrated significant change (*p* = 0.001) from baseline to post-treatment in the CPAP group as compared to controls [2]. Importantly, this domain was the only sub-domain on the K-MMSE that independently demonstrated statistically significant change [2]. Aaronson et al. analyzed the results of three cognitive tasks that evaluated vigilance and attention (Psychomotor Vigilance Task, d2 Test of Attention, and the Color Trails Test) and compiled the results to create a combined analysis of these cognitive domains [3]. Compared to the control group, the CPAP group did not show significantly greater improvement in the domain of vigilance following treatment (*p* = 0.34) but did show significantly greater improvement in the domain of attention (*p* = 0.048) with a large effect size (ηp^2^ = 0.09) [3].

The investigation by Ryan et al. measured vigilance using the sustained attention to response task (SART) as well as the digit and visual spatial span-forwards task [6]. No significant within-group or between-group recovery was demonstrated in this area [6].

#### 3.4.3. Executive Function

All included studies reported outcome measures related to executive functioning. Kim et al. measured temporal organization using the orientation to time task on the K-MMSE [2]. This score did not show a significant between-group change pre- to post-CPAP treatment [2]. Similarly, in the study by Ryan et al. there was not a significant between-group difference on a task related to executive function (digit span and visual spatial span-backward task) [6]. However, the CPAP group demonstrated significant within-group improvement on this task (*p* = 0.03), which was not observed in the control group [6].

Aaronson and colleagues, on the other hand, did find significant between-group differences in executive function task performance. When comparing the treatment and control groups scores on two tasks (The D-KEFS Trail Making Test and the Tower of London Task), executive function outcomes showed a significantly larger improvement in the CPAP group following treatment (*p* = 0.001), with a large partial effect size (ηp^2^ = 0.26) [3].

#### 3.4.4. Memory

Two of the included studies reported scores in the domain of memory, and neither reported statistically significant between-group improvement in this area. In the study by Kim and colleagues, the K-MMSE scores for the registration and recall domains did not show statistically significant differences between groups (*p* = 0.554 and 0.558, respectively) [2]. Aaronson et al. utilized 5 tests (Rey’s Auditory Verbal Learning Test, the WAIS-III Letter Number Sequencing, Category Fluency, the Location Learning Test, and the WMS-IV Symbol Span) to measure outcomes of general memory and working memory [3]. The results indicated that there was no statistically significant improvement in general memory or working memory following the intervention period (*p* = 0.32 and *p* = 0.16, respectively) [3].

### 3.5. Primary Outcome Measure: Language

No statistically significant between-group improvements were demonstrated in the included studies in the domain of language [2,3,6]. To compare between-group outcomes, Kim et al. evaluated language using the K-MMSE (*p* = 0.501), Aaronson et al. used a variety of measures to compile a language domain score (*p* = 0.11), and Ryan et al. administered the Cognitive FIM, which required participants to use cognitive resources and abilities needed to effectively use language (*p* = 0.76) [2,3,6]. While no between-group significant results were reported, Ryan and colleagues reported that the CPAP group showed significant within-group improvement on the Cognitive FIM following treatment (*p* = 0.03) [6].

### 3.6. Secondary Outcome Measure: Stroke Severity and Functional Status

Stroke severity and functional status were included as outcomes of interest in each of the included studies. The study by Kim et al. measured level of impairment due to stroke using the Korean version of the NIHSS [2]. Aaronson and colleagues also used the NIHSS combined with the CNS to generate one functional outcome score [3]. After intervention, the CPAP group did not show significant improvement in stroke severity or functional level compared to the control group in either of the above studies (*p* = 0.157 and *p* = 0.08, respectively) [2,3].

Ryan and colleagues also used the CNS to evaluate overall neurological and functional status [6]. Compared to the control group, the CPAP group demonstrated significantly greater improvement on this measure (*p* < 0.001). This study additionally reported the total score on the Functional Independence Measure (FIM), which showed significant within-group (CPAP *p* < 0.001; control *p* < 0.001) improvements in functional status following the intervention period, but no significant between-group difference in improvements (*p* = 0.07) [6].

### 3.7. Secondary Outcome Measure: Sleepiness

The included RCTs used a variety of the following measures to evaluate sleepiness and sleep quality: the Epworth Sleepiness Scale (ESS), AHI, the Sleep Quality Scale (SQS) and the Stanford Sleepiness Scale (SSS). Two of the three studies showed significantly greater improvements in scores on the ESS (*p* = 0.003, *p* < 0.0001) as well as decreased AHI (*p* = 0.001, *p* < 0.0001) [2,3,6]. In addition, one of the two investigations to use the SSS demonstrated statistically significant between-group improvements (*p* = 0.005) [6]. However, the other study, by Aaronson et al., did not (*p* = 0.41) [3]. Aaronson et al. also did not report a significant change in scores on the SQS [3].

### 3.8. Secondary Outcome Measure: Affect Status

All included studies reported affect status as one of their outcome measures; however, results varied. The investigation by Kim et al. used the EuroQol 5-Dimension questionnaire to assess the quality of life of the participants, which includes a domain for anxiety/depression [2]. However, the anxiety/depression domain score was not included in the reported scores. After treatment, there was not a significant within-group or between-group difference in overall EuroQoL scores [2]. Similarly, Aaronson and colleagues did not report significant between-group differences in anxiety or depression scores following the intervention period using the Hospital Anxiety and Depression Scale (*p* = 0.45 and *p* = 0.33, respectively) [3].

However, one of the included studies did report some positive changes in affect status following CPAP treatment. The Beck Depression Inventory (BDI) was used by Ryan and colleagues to report a total score, as well as affective and somatic component scores [6]. There was no significant improvement in the total BDI score following intervention in the control group or the CPAP group (*p* = 0.35 and *p* = 0.76, respectively); however, there was a significant decrease (positive change) in the CPAP group’s affective component score when compared to the control group (*p* = 0.006) [6].

## 4. Discussion

Findings from this review of the literature do not reveal clear conclusions about the benefits of CPAP on cognitive functioning or language in stroke patients. The use of CPAP treatment may lessen cognitive decline exacerbated by SDB by improving functions that can specifically be affected in SDB patients, such as attention and executive functioning [7,8]. Therefore, the presence of a co-occurring SDB diagnosis with a stroke or TIA diagnosis may account for some of the benefits experienced following CPAP treatment in these domains. This is one of several factors and limitations to consider when interpreting current results in the literature.

There is a large degree of interaction between cognition, psychological state, and physiological state. Therefore, it is important to note that although research studies use assessments to evaluate specific components, results can be applied more holistically. For example, adequate attention is necessary to complete a working memory task, as are the executive functions of planning, organizing, reasoning, and mental flexibility. These skills are also critical for effective use of language. Because of this synergy, results should be interpreted comprehensively when possible.

### 4.1. Primary Outcomes

Cognition is defined as the process of gathering, comprehending, and recalling information to apply it appropriately. Communication and collaboration across the different domains of cognition are essential for proper functioning of the whole system. Sleep is necessary for the integration of cognitive encoding, consideration, storage, retrieval, and application of information. SDB causes sleep fragmentation, which has a negative effect on neurocognitive recovery after stroke [9,10,11,12]. Regarding overall measures of cognition, the correlation of change in K-MMSE scores, ESS scores, and AHI from baseline to post-treatment in the study by Kim et al. supports the idea of a relationship between uninterrupted sleep and cognitive performance [2,13].

While overall measures of cognition are important to consider for gaining a holistic view of cognitive recovery, it is also valuable to evaluate individual domains of cognition to determine specific treatment benefits. Vigilance and attention are often used synonymously to refer to alertness, the ability to take in and prune away unnecessary information, the ability to understand presented stimuli, and the ability to sustain attention. Aaronson et al. and Kim et al. found CPAP treatment resulted in significant improvements in attention, while Ryan et al. found no within-group or between-group benefits on vigilance [2,3,6]. Executive functioning is essential for the cognitive input of information, as it encompasses abilities required to process information, including planning, organizing, being able to add or change information based on new learning, and making judgements based on knowledge. The studies by Kim et al. and Ryan et al. did not report significant between-group differences in outcomes on tasks related to executive function [2,6]. Aaronson and colleagues, however, did find significant between-group differences in executive function task performance, once again demonstrating discrepancies in results between studies [3]. Memory further integrates attention into the consolidation of information being presented by assisting in processing and retaining that information. No studies included in this review found statistically significant between-group differences in memory from baseline to post-treatment measures. These varied results in the domains of cognition make it difficult to draw conclusions from this review.

Effective language skills require use of all previously mentioned cognitive domains. Language requires vigilance and attention to take in the information as well as memory to briefly hold the information. The executive functions involved in language include: planning, organizing information, being mentally flexible in interpretation and judgements, and reasoning to relate messages to personally developed schemas in order to make decisions. While the included studies reported varied results in these different cognitive domains, none of the included studies reported statistically significant between-group differences in the language domain. However, Ryan and colleagues reported that the CPAP group showed significant within-group improvement on the Cognitive FIM following treatment (*p* = 0.03), which required participants to use many cognitive functions necessary for language [6]. Given the observed improvements in cognitive domains known to contribute to language, but lack of direct improvement in language itself, further research is necessary to provide conclusive information.

### 4.2. Secondary Outcomes

The diversity among types of strokes, severity of damage, severity of stroke outcomes, and personal factors, such as age and co-occurring conditions, are all factors that play a large role in individual stroke recovery. Of the three studies included in this review, only one reported between-group significant improvement in functional status and stroke severity following CPAP treatment [6]. Therefore, further analysis of these related factors is necessary to draw conclusions on benefits on functional status from CPAP treatment.

Sleep has proven to be vital to optimal body and brain health. Some specific areas that can be affected by sleep include, but are not limited to, attention, learning, memory, reasoning, decision making, and language [14,15]. Sleep impacts these functional areas, as proper sleep improves focus and aids in the mental acquisition and consolidation of information [14,15]. Two of the included studies reported significant improvements in subjective and objective measures of sleep quality following CPAP treatment [2,6]. However, the study by Aaronson et al. did not report significant improvements on measures of sleep quality [3].

Anxiety and depression can impact brain function, including the ability to organize and interpret information and to adapt to change. Approximately 25–40% of post-stroke patients develop depression, which decreases the likelihood of optimal neurological rehabilitation [16,17]. Therefore, effective treatment of symptoms of depression in post-stroke patients could potentially facilitate neurological recovery. All the included studies reported affect status, including anxiety and depression, in their outcome measures. One of the three studies demonstrated significant between-group improvements on these measures, while two did not [2,3,6].

### 4.3. Limitations of the Included Studies

While the included studies met our inclusion criteria and offered valuable results regarding our outcome measures of interest, they were not without their limitations. All the included studies had a small sample size (N < 50) [2,3,6]. In addition, all studies used a short intervention period (4 weeks or less) of CPAP, which may not have been long enough to determine the extent of possible recovery. When investigating treatment duration, it is important to factor in changes that can be experienced post-stroke as well as the amount of progress that is possible in treatment and rehabilitation [18]. Two of the three included studies had CPAP groups that met criteria for adequate CPAP compliance; however, the study by Kim et al. did not accurately measure compliance [2,3,6]. Therefore, we can conclude that only one included study, by Ryan et al., truly demonstrated CPAP compliance.

All the included studies took place while participants were in inpatient rehabilitation units [2,3,6]. This homogeneity in the included studies has pros and cons. The reported outcomes of use of CPAP in this setting could lessen the applicability of using CPAP outside of the inpatient setting, as family and caregivers would administer CPAP at home rather than trained professionals. However, the apparent benefit of good compliance due to the motivation and constant presence of medical professionals may have provided more substantial benefit than if the treatment was not adequately completed. Moreover, starting CPAP intervention early in the acute phase after stroke may extend the life of the penumbra, the area around the site of the stroke, which would encourage blood flow and oxygen transport [17]. The immediate benefits of this effect, including improved cognition, decreased drowsiness, and decreased depression, could potentially have a further positive influence on recovery after stroke [19].

Kim et al.‘s study had several limitations, including a shorter intervention period of 3 weeks, due to a limited hospitalization period, and broad inclusion criteria [2]^.^ The investigators did not determine if patients had previously undiagnosed sleep apnea prior to their stroke; therefore, unreported prior sleep apnea may have caused some worsened symptoms at baseline, which subsequently showed more improvement [2]. Additionally, patients with severe strokes and severe SDB did not experience any benefits from the CPAP treatment, which may have skewed outcome measure scores [2]. Aaronson et al. completed a comprehensive neuropsychological battery to assess cognitive outcomes, while the other included studies used brief cognitive measures [3]. However, some patients in their study demonstrated high levels of neurological function at baseline, so further improvement was not possible [3]. Finally, in the study by Ryan et al., the functional measures utilized (NIHSS and CNS) demonstrated robust ceiling effects during post-treatment testing [6].

Finally, the included studies either did not sufficiently differentiate language from other communication and cognitive impairments or did not provide enough initial testing information about the patient groups to determine if language was an area that needed improvement. Although all language assessments inherently require some level of intact cognitive functioning to complete, the measures used in the studies included in this review often combined measures of language and cognition. For example, the Cognitive FIM overall score (reported in the study by Ryan and colleagues) contains five tasks, two that assess communication (comprehension, expression) and three that assess social cognition (social interaction, problem solving, memory) [6]. Given that the overall reported score reflects all these subdomains, it is impossible to determine from this measure alone whether communication difficulties or subsequent change in these measures arose from language or from other communicative domains (e.g., speech or cognition). An overall score that combines domains could potentially mask change in individual underlying subareas. For example, it is possible that the communication domain improved post-treatment, but these effects were masked by lack of change in the cognitive domain. Kim and colleagues reported both overall K-MMSE score as well as language domain sub score, which did allow for separate assessment of language and cognition [2]. However, initial language function of the patient groups was unspecified, so it could not be determined if language change would be expected (i.e., if few patients showed language difficulty pre-treatment, we would not expect to see significant change after treatment). Aaronson and colleagues used a category fluency task to assess language, but again it was not clear if the patient groups had an initial language impairment to improve [3]. Given the limitations of these measures and reported scores, it is difficult to draw definitive conclusions regarding the effects of CPAP on language in the post-stroke population.

### 4.4. Future Directions of Related Research

While this review offers inclusive results regarding the use CPAP and post-stroke cognitive recovery, more research is needed to come to conclusions regarding its effects on language. More sensitive and comprehensive assessments, larger sample sizes, and longer treatment durations are necessary. Additionally, studies that investigate specific stroke populations, (i.e., subtypes, co-occurring conditions, severity of impairment, etc.) would be conducive to determining more specific outcomes of CPAP treatment. Finally, methods to improve and maintain CPAP compliance would be of interest in future studies, as compliance has a direct impact on outcomes.

## 5. Conclusions

Obstructive sleep apnea can affect cognitive, neurological, and functional skills, especially in the post-stroke population. The beneficial effects found for CPAP treatment suggest that this intervention has the potential be considered as a part of post-stroke patients’ comprehensive rehabilitation plan; however, further research is needed to draw conclusions. Although this review of the literature found CPAP to have varying effects on improving cognitive domains which contribute to language, there was not significant recovery in the domain of language itself. Further research is needed to determine the potential effects of CPAP treatment on speech and language recovery among stroke patients.

## Figures and Tables

**Figure 1 brainsci-12-00379-f001:**
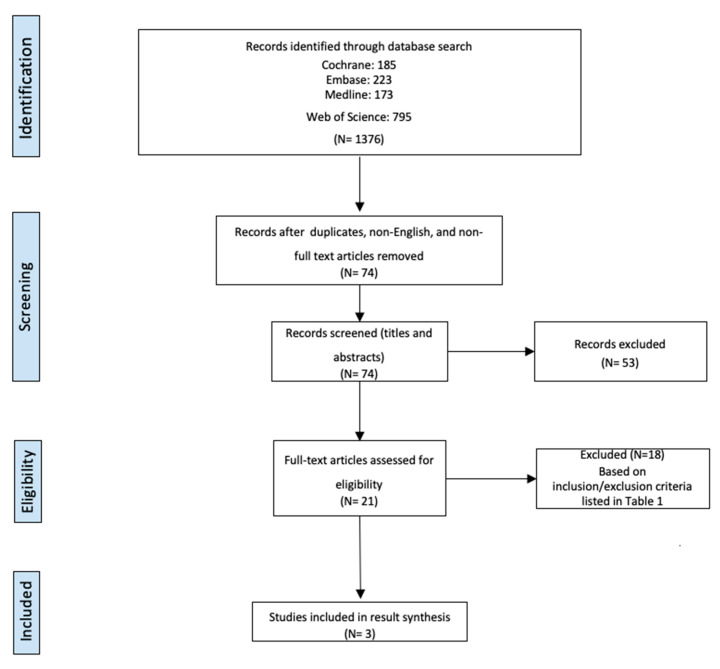
Flowchart of study retrieval.

**Table 1 brainsci-12-00379-t001:** Inclusion and Exclusion Criteria.

Inclusion	Exclusion
1. Published between 2011–2021	1. BiPAP
2. Randomized controlled trials (RCTs)	2. EPAP
3. CPAP intervention compared to standard care or sham CPAP/control (nCPAP and aCPAP included)	3. Cardiovascular topics of interest
4. Adult patients (18 years +) with SDB	4. Oral/Dental Appliances to treat sleep apnea
5. Patient experienced a stroke or TIA (transient ischemic attack)	5. Articles unable to acquire through inter-library loan
6. Patient has SDB including, but not limited to, OSA	6. Focus of adherence to CPAP
7. Baseline and post-tx measures reported	7. Relevant baseline measure but no post-tx measure, or vice versa
8. Outcome measures related to general neurocognition or language	8. Duplicate publications
	9. Motor measures of functioning

**Table 2 brainsci-12-00379-t002:** Characteristics of Included Articles.

Study	Participants, n	Sham or Control Group	Type of Stroke or TIA	Onset of Treatment	Duration of Treatment	Follow-Up	Relevant Measures	Mean CPAP Use/Adherence
Kim et al., 2019	CPAP = 20 Control = 20 Total = 40	Control	Cerebral Infarction or Hemorrhage	Between 7 days and 6 months post-stroke 4.6 +/− 2.8 days after a person with stroke was admitted	3 weeks	T0 = Baseline T1 = 3 weeks	**Cognitive:** 1. Korean Version of Mini-Mental State Exam *Domains = Orientation to time, orientation to place, registration, attention and calculation, recall, language, drawing* **Mental Health:** 1. EuroQol-5 Dimension (EQ-5D) **Sleep:** 1. Epworth Sleepiness Scale (ESS) **NeuroFunctional Status/Stroke Severity:** 1. Korean version of NIHSS	“Highly compliant” = More than 4 h/day for 5 or more days/week ** Not accurately measured >> accuracy of the reported CPAP compliance could not be verified, as these data were only collected from the CPAP machines themselves, which are not a reliable measure of true treatment compliance.
Aaronson et al., 2016	CPAP = 20 Control = 16 Total = 36	Control	Ischemic or Hemorrhagic	Between 1 and 16 weeks post-stroke	4 weeks ** Control group was offered CPAP treatment at the end of the 4 weeks **	T0 = Baseline T1 = 4 weeks of intervention T2 = 2 months	**Cognitive:** 1. Psychomotor Vigilance *(Organizing movement and sustaining attention in response to visual stimuli)* 2. * D-KEFS Trail Making Test (*Temporal sequencing and mental flexibility)* 3. d2 Test of Attention *(Processing speed, rule compliance, and quality of performance; sustained and selective attention)* 4. * Rey’s Auditory Verbal Learning Test *(Verbal memory)* 5. WAIS-III Letter Number Sequencing *(Working memory)* 6. Tower of London *(Executive functioning and planning)* 7. * Category Fluency *(Verbal fluency and semantic memory)* 8. Bell’s Task *(Visuoperception and neglect)* 9. Finger Tapping test *(Psychomotor ability and motor speed)* 10. WAIS-III Matrix Reasoning *(Perceptual organization; nonverbal abstract reasoning)* 11. **Color Trails Test *(Sustained attention and sequencing)* 12. ** Location Learning Test *(Visuospatial memory)* 13. ** WMS-IV Symbol span *(Visual working memory)* ** = Language task* *** = Non-Verbal Alternatives for PWA* **Mental Health:** 1. Hospital Anxiety and Depression Scale (HADS) **Sleep:** 1. Stanford Sleepiness Scale (SSS) 2. Sleep Quality Scale (SQS) **NeuroFunctional Status/Stroke Severity:** 1. National Institutes of Health Stroke Scale (NIHSS) 2. Canadian Neurological Scale (CNS)	“Compliant” = 1h/night “Good” = More than 4 h for 5 or more nights/week T1 = 2.5 h/night (+/− 2.8; range 0–9) ~9 CPAP tx participants were noncompliant ~11 complaint CPAP tx particpants mean 4.4 h/night (+/− 2.5; range 1.3–9) ~7 particpants had good compliance ~10/14 control participants started CPAP treatment at this point with 3.2 h/night (+/− 2.5; range 0.3–7.8) T2 = 4.9 h/night (+/− 2.9) ~8/11 compliant CPAP tx participants were still using the CPAP
Ryan et al., 2011	CPAP = 22 Control = 22 Total = 44	Control	Ischemic or Hemorrhagic	CPAP Group = 21.5 +/− 8.7 days post-stroke Control Group = 19.7 +/− 16.8 days post-stroke	4 weeks	T0 = Baseline T1 = 4 weeks	**Cognitive:** 1. Sustained Attention Response Task (SART) and Digit or Spatial Span-Forwards *[Vigilance]* 2. Digit or Spatial Span Backwards *[Executive functioning]* 3. Functional Independence Measure (FIM) Cognition *[Communication (auditory and/or visual comprehension & vocal and/or nonvocal expression) & social cognition (social interaction, problem solving, memory)]* 4. Canadian Neurological Scale (CNS) Cognition **Functional:** 1. Functional Outcome Measure (FIM) Total Score 2. Canadian Neurological Scale (CNS) Total Score **Mental Health:** 1. Beck Depression Inventory (BDI) **Sleep:** 1. Epworth Sleepiness Scale (ESS) 2. Stanford Sleepiness Scale (SSS)	4.96 +/− 2.25 h/day—excellent compliance

## Data Availability

Not applicable.

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
