# Peer review of "Effects of CPAP on Language Recovery in Post-Stroke Aphasia: A Review of Recent Literature"

_brainsci, 2022, doi:10.3390/brainsci12030379_

Round 1

Reviewer 1 Report

Thank you for addressing this interesting topic. This comment is just to add to the other complete reviews as supplementary feedback to only one aspect.

The title of the review suggests that statements can be made about the influence of CPAP on language recovery in aphasia. The measures of language discussed in the three articles of the review are only insufficiently suitable to actually measure original linguistic functions. For example, the FIM, which is used in one study, measures communication in a relatively undifferentiated way, but the score does not allow any conclusions to be drawn about the origin of the communication disorder - aphasia, dysarthria, or cognitive communication disorder. Also, language subtests from dementia tests used in another study are poorly suited to differentiate aphasic disorders from dementia-induced language disorders. The word fluency task used by Aaronson et al. does measure word retrieval, but again, it is not possible to infer an underlying aphasic disorder, as word fluency tasks in particular also rely on good executive functioning. This should be considered more in the discussion or the title should be adjusted.

Author Response

We would like to thank you for taking the time to read our manuscript and provide thoughtful and helpful feedback. We have provided each reviewer comment below with bold text used to indicate our response.

Thank you for addressing this interesting topic. This comment is just to add to the other complete reviews as supplementary feedback to only one aspect.

Thank you for your positive feedback on our choice of topic. We appreciate your thanks.

The title of the review suggests that statements can be made about the influence of CPAP on language recovery in aphasia. The measures of language discussed in the three articles of the review are only insufficiently suitable to actually measure original linguistic functions. For example, the FIM, which is used in one study, measures communication in a relatively undifferentiated way, but the score does not allow any conclusions to be drawn about the origin of the communication disorder - aphasia, dysarthria, or cognitive communication disorder. Also, language subtests from dementia tests used in another study are poorly suited to differentiate aphasic disorders from dementia-induced language disorders. The word fluency task used by Aaronson et al. does measure word retrieval, but again, it is not possible to infer an underlying aphasic disorder, as word fluency tasks in particular also rely on good executive functioning. This should be considered more in the discussion or the title should be adjusted.

Thank you for this thoughtful insight. We have taken your comment into consideration and added to the Limitations section of the discussion to include this shortcoming of the included studies. We have also edited the Abstract to include our results, so that readers can gain an understanding of the findings from looking at the title and abstract.

Reviewer 2 Report

Lines 86 & 92 -Define nCPAP, aCPAP, biPAP, EPAP

Lines 98- It is mentioned that 2 articles were unable to be accessed through inter-library loan and were excluded. Citations should be included for these two articles in the manuscript text.

Lines 119-120. It is mentioned that three RCTS were included but “descriptive data from each of these four studies” are presented. Please correct or clarify.

Lines 126-129.  As written Kim and colleagues used a 3 week period for CPAP treatment, while the other included studies used an intervention period of 4 weeks”  Shouldn't this be “while the remaining two studies used an intervention period of 4 weeks”?

Line 375  Shouldn’t this say “inclusive” rather than “potentially promising” as the authors mention earlier that it is difficult to arrive at conclusions regarding CPAP based on these 3 studies.

Line385-387.  This sentence is somewhat misleading, as more evidence would be needed than presented would be needed to support such a statement. This statement should be re-phrased as definitive conclusions can’t be drawn, particularly when only 1 study accurately measured CPAP compliance.

Author Response

We would like to thank you for taking the time to read our manuscript and provide thoughtful and helpful feedback. We have provided each reviewer comment below with bold text used to indicate our response.

Lines 86 & 92 -Define nCPAP, aCPAP, biPAP, EPAP

Thank you for this suggestion. We have defined each term to provide specific information about the different respiratory treatments discussed.

Lines 98- It is mentioned that 2 articles were unable to be accessed through inter-library loan and were excluded. Citations should be included for these two articles in the manuscript text.

Thank you for this suggestion. We have added the requested citations to the manuscript text.

Lines 119-120. It is mentioned that three RCTS were included but “descriptive data from each of these four studies” are presented. Please correct or clarify.

Thank you for pointing out this error. Only three studies were included, and this has been corrected.

Lines 126-129.  As written Kim and colleagues used a 3 week period for CPAP treatment, while the other included studies used an intervention period of 4 weeks”  Shouldn't this be “while the remaining two studies used an intervention period of 4 weeks”?

Thank you for pointing out this error and offering a helpful suggestion to change our wording. We have taken the suggestion and modified this sentence as you detailed in your comment.

Line 375  Shouldn’t this say “inclusive” rather than “potentially promising” as the authors mention earlier that it is difficult to arrive at conclusions regarding CPAP based on these 3 studies.

Thank you for this suggestion. We have changed the language from “potentially promising” to “inclusive” based on your comment. We agree that it is difficult to draw conclusions based on the included studies, and this language better summarizes our findings than what we had originally written.

Line385-387.  This sentence is somewhat misleading, as more evidence would be needed than presented would be needed to support such a statement. This statement should be re-phrased as definitive conclusions can’t be drawn, particularly when only 1 study accurately measured CPAP compliance.

Thank you for your comment. You are correct that definitive conclusions cannot be drawn from this review. We have changed the language in this conclusion section to better reflect the scope of the review and conclusions that can be drawn from the analyzed studies.

Reviewer 3 Report

Major comments:

  1. It is not clear how the authors reach N = 74 after the identification stage. Assuming only the duplicate records were removed and no exclusion criterion was applied then the authors should achieve at least N = 173, i.e., the minimum number of articles reported by the Medline database. Please explain this discrepancy.
  2. The review concludes in line 385 that "The beneficial effects found for CPAP treatment suggest that this intervention should be considered as a part of post-stroke patients’ comprehensive rehabilitation plan.". The objective pieces of evidence described in the reviewed studies are, however, inconsistent. The statement should be corrected.

Author Response

We would like to thank you for taking the time to read our manuscript and provide thoughtful and helpful feedback. We have provided each reviewer comment below with bold text used to indicate our response.

It is not clear how the authors reach N = 74 after the identification stage. Assuming only the duplicate records were removed and no exclusion criterion was applied then the authors should achieve at least N = 173, i.e., the minimum number of articles reported by the Medline database. Please explain this discrepancy.

Thank you for your comment and for pointing out this discrepancy. We have added to the Methods section and Figure 1 to address this. Namely, that duplicates were not only removed from across databases, but within the same database across different queries. For example, the first two search queries completed in Medline were: (1) TOPIC: (stroke) AND TOPIC: (continuous positive airway pressure) and (2) TOPIC: (stroke) AND TOPIC: (continuous positive airway pressure) AND TOPIC: (obstructive sleep apnea). These two search queries would lead to many duplicate articles being identified within the same database. We also clarified that non-English articles and non-full-text articles (e.g., abstracts, posters, presentations) were removed at this stage.

The review concludes in line 385 that "The beneficial effects found for CPAP treatment suggest that this intervention should be considered as a part of post-stroke patients’ comprehensive rehabilitation plan.". The objective pieces of evidence described in the reviewed studies are, however, inconsistent. The statement should be corrected.

Thank you for your comment. You are correct that definitive conclusions cannot be drawn from this review. We have changed the language in this conclusion section to better reflect the scope of the review and conclusions that can be drawn from the analyzed studies.

Round 2

Reviewer 1 Report

Thank you for editing and integrating all the comments and modifying the abstract. For my feeling as a non-native speaker, the English should be carefully checked again (e.g. if/whether).